# *Microplitis bicoloratus* Bracovirus Promotes Cyclophilin D-Acetylation at Lysine 125 That Correlates with Apoptosis during Insect Immunosuppression

**DOI:** 10.3390/v15071491

**Published:** 2023-06-30

**Authors:** Dan Yu, Pan Zhang, Cuixian Xu, Yan Hu, Yaping Liang, Ming Li

**Affiliations:** 1Yunnan International Joint Laboratory of Virology & Immunology, School of Life Sciences, Yunnan University, Kunming 650500, China; 20210172@ynu.edu.cn (D.Y.); yndxxb@ynu.edu.cn (P.Z.); xucuixian@mail.xingaojiao.com (C.X.); huyan@walvax.com (Y.H.); liangyaping@mail.ynu.edu.cn (Y.L.); 2Yunnan Provincial Medical Investment Management Group Co., Ltd., Kunming 650500, China; 3School of Health, Yunnan Technology and Business University, Kunming 650500, China

**Keywords:** *Microplitis bicoloratus* bracovirus (MbBV), cyclophilin D (CypD), acetylation, apoptosis, immunosuppression

## Abstract

Cyclophilin D (CypD) is regulated during the innate immune response of insects. However, the mechanism by which CypD is activated under innate immunosuppression is not understood. *Microplitis bicoloratus* bracovirus (MbBV), a symbiotic virus in the parasitoid wasp, *Microplitis bicoloratus*, suppresses innate immunity in parasitized *Spodoptera litura*. Here, we demonstrate that MbBV promotes the CypD acetylation of *S. litura*, resulting in an immunosuppressive phenotype characterized by increased apoptosis of hemocytes and MbBV-infected cells. Under MbBV infection, the inhibition of CypD acetylation significantly rescued the apoptotic cells induced by MbBV, and the point-mutant fusion proteins of CypDK125R-V5 were deacetylated. The CypD-V5 fusion proteins were acetylated in MbBV-infected cells. Deacetylation of CypDK125R-V5 can also suppress the MbBV-induced increase in apoptosis. These results indicate that CypD is involved in the MbBV-suppressed innate immune response via the CypD-acetylation pathway and *S. litura* CypD is acetylated on K125.

## 1. Introduction

*Microplitis bicoloratus* bracovirus (MbBVs) is a polydnavirus (PDV). PDVs are part of the parasitoid/polydnavirus/host system and are involved in altering host gene expression to induce immunosuppression [1,2]. *Microplitis bicoloratus* (Hymenoptera: Braconidae) is an endoparasitoid of *Spodoptera litura*, a serious agricultural pest. Female wasps carry MbBV particles in the lumen of the calyx and oviduct [3,4]. Female wasps inject virus particles along with their eggs into the hemocoel of larval *S. litura*. The viral genome is then integrated into the host chromosomes to express viral genes that suppress the host immune response and ensure the survival and development of the wasp eggs [3,4,5,6,7]. Although bracoviruses play key roles in inhibiting host immune responses by inducing hemocyte apoptosis, the details of this process are unclear.

Mitochondria are the energy center of cells and are involved in ATP production, Ca^2+^ regulation, neurotransmitter synthesis and degradation, ROS production, apoptosis, and metabolic regulation [8]. Many viruses can modulate mitochondrial apoptosis of infected cells by altering the mitochondrial permeability transition pore (mPTP). The mPTP is a non-selective channel with a diameter of about 2–3 nm, which reaches across both the outer membrane and the inner membrane and allows the free passage of molecules with molecular weights of less than 1.5 kDa [9]. mPTP regulation is performed by the voltage-dependent anion channel (VDAC) in the outer membrane, adenine nucleotide translocase (ANT) in the inner membrane, and cyclophilin D (CypD) in the mitochondrial matrix [10]. CypD is a member of the cyclophilin family that functions as a mitochondrial protein chaperone and plays an important role in mitochondrial biology [11]. CypD initiates the assembly and opening of the mPTP. The irreversible opening of the mPTP culminates in the loss of mitochondrial transmembrane potential (∆Ψ_m_), an arrest of mitochondrial bioenergetic and biosynthetic functions, and the release of mitochondrial intermembrane space (IMS) proteins into the cytosol [12,13,14,15,16]. More studies have shown that CypD binds to ANT1 to form a complex that promotes mPTP opening [15,17,18,19]. This means that CypD may be involved in the regulation of cell death in the mitochondrial pathway.

Protein acetylation is a reversible post-translational modification (PTM), which adds an acetyl moiety to the epsilon-amino group of lysine residues [20,21,22]. Lysine acetylation plays a critical role in histone modification and influences gene expression [23]. The acetylation of mitochondrial protein also occurs [24]. Acetylated CypD is considered to be its active form, and its acetylation levels are increased under stress conditions such as hypoxia/reoxygenation, heart failure, and drug treatment [25,26,27,28]. The acetylation of CypD on lysine166 in mice (K166, homologous to K167 in human CypD) promotes cell death by sensitizing the mPTP [16,29]. There are no studies on the acetylation of CypD and its correlation with immunity and apoptosis in insects.

Antiviral innate immunity is conserved among mammals and invertebrates [30]. The innate immune response of insects includes both humoral and cellular immunity, which act jointly to resist external adverse stimuli such as viral infection [31,32,33]. In the *M. bicoloratus*–MbBV–*S. litura* parasitoid system, transcriptome analysis has revealed that *M. bicoloratus* parasitism can regulate critical signaling pathways of host hemocytes to promote apoptosis that suppresses host cellular immunity [34]. We previously found that *M. bicoloratus* parasitism upregulates the expression level of CypD and promotes host hemocyte apoptosis via CypD–p53 interaction during immune suppression [34,35]. However, the molecular mechanism of CypD activation is unclear.

In this study, we determined if MbBV promotes CypD acetylation, its active form, to participate in the induction of host hemocyte apoptosis during MbBV-mediated immunosuppression. We conducted molecular and cellular experiments to study this possibility and to increase our understanding of the molecular mechanisms underlying MbBV-mediated apoptosis.

## 2. Materials and Methods

### 2.1. Insect Rearing

*S. litura* larvae were reared on an artificial diet as previously described by Li et al. [36]. Rearing conditions were 27 °C ± 1 °C, relative humidity (RH) 60–80%, and a 12:12 h (L:D) photoperiod. *M. bicoloratus* parasitoids were maintained on laboratory-reared *S. litura* larvae, and the adults were supplied with honey as a dietary supplement. The parasitoid colony was passaged as previously described [3].

### 2.2. Isolation of Hemocytes from S. litura Larvae

Hemocyte samples were isolated from parasitized 2nd instar *S. litura* larvae at 6 d post-parasitization (p.p.) or 10-d p.p. [3]. The samples were centrifuged for 5 min at 1000× *g*, and the resulting pellets were collected as parasitized hemocytes (P-hemocytes). We also collected hemocytes from un-parasitized 4th instar *S. litura* larvae at 6 d to serve as unparasitized controls (NP).

### 2.3. Cell Culture

Spli221 (TUAT-Spli221) adherent cells (Thermo Fisher Scientific, Shanghai, China) were derived from the pupal ovaries of *S. litura* [37]. The cells were cultured in the TNM-FH insect culture medium with 10% fetal bovine serum (FBS) (HyClone, GE Life Sciences, Beijing, China). The cultures were maintained at 27 °C and passaged in 25-cm^2^ tissue culture flasks (Corning, Corning, NY, USA).

### 2.4. Western Blotting

Western blotting was performed as previously described [38]. Briefly, 50 µg of protein was loaded per sample. Total protein was isolated and analyzed by blotting using appropriate antibodies: mouse anti-Tubulin monoclonal antibody (1:2000; #AT819; Beyotime, Shanghai, China), anti-Ac-Lys antibody (1:2000; #ICP0380; ImmuneChem Pharmaceuticals); HRP-labeled goat anti-mouse IgG (H + L) (1:2000; #A0216; Beyotime, Shanghai, China); HRP-labeled goat anti-rabbit IgG (H + L) (1:2000; #A0208; Beyotime, Shanghai, China), anti-mouse V5 Tag monoclonal antibody (1:2000; #R96025; Invitrogen, CA, USA), goat anti-mouse horseradish peroxidase-conjugated secondary antibody (1:2000; #0216; Beyotime, Shanghai, China), anti-CypD antibody, which was a rabbit polyclonal antibody generated for CypD, cloned from *M. bicoloratus* hemocytes and expressed in *Escherichia coli* using the pET-32a-CypD plasmid (1:2000; Bioworld Technology, Inc., Nanjing, China). ImageJ (National Institutes of Health, Bethesda, MD, USA) was used to measure protein band density.

### 2.5. Isolation and Purification of MbBV Particles and Infection of Spli221 Cells

Viral particles were purified using a published protocol [33]. In detail, 50 fresh adult *M. bicoloratus* were frozen at −20 °C for 10 min and then put on ice. The reproductive tracts of female wasps were excised under a binocular stereomicroscope, and the separated ovaries were collected into 1.5 mL Eppendorf tubes on ice. The calyces were punctured using forceps. The exuded calyx contents were collected in 1× phosphate-buffered saline (PBS, pH 7.4) using a 2.5-mL syringe. The mixture was centrifuged for 3 min at 960× *g*, 4 °C to remove eggs and cellular debris. A 0.45-μm syringe filter was used to purify the viral particles. Spli221 cells were seeded in a six-well culture plate at a density of 2.5 × 10^6^ cells per well and cultured for 2 h. The experiments were expressed in wasp equivalents. One wasp equivalent in 20 μL of cell culture medium was inoculated per 1 × 10^5^ Spli221 cells.

### 2.6. Inhibition of CypD Acetylation

Spli221 cells were seeded in a six-well culture plate at a density of 5 × 10^5^ cells per well. After cell adhesion, the cells were treated with 1 μM of cyclosporine A (CsA) (#12088; Cayman Chemical, MI, USA) or 1 μM of polydatin (#A10741; Adooq Bioscience) and cultured at 27 °C for 24 h. The cells were infected with MbBV for an additional 24 h. The experiments were expressed in wasp equivalents. One wasp equivalent in 20 μL of cell culture medium was inoculated per 5 × 10^5^ Spli221 cells, and Western blotting was performed as described in Section 2.4.

### 2.7. Total RNA Isolation from Hemocytes and cDNA Synthesis

Hemocytes were isolated from *S. litura* larvae at 6–10 d post-parasitization (p.p.). Unparasitized larvae at the same age were used as the control group. The isolation of total RNA and subsequent cDNA synthesis were performed using previously described protocols [34].

### 2.8. Plasmid Construction and Expression

CypD was amplified by PCR using *S. litura* cDNA as a template. Primers containing EcoRI and NotI sites (underlined) were used and are as follows: pIZT/V5-His-CypD (pIZT-CypD)-F (5′-GAATTCATGGGTTTACCGCGTGTT-3′) and pIZT/V5-His-CypD (pIZT-CypD)-R (5′-GCGGCCGCACAATTGTCCAC-3′). The genes were directionally cloned into a pMD-19-T vector (Takara, Dalian, China), and the insert was confirmed by sequencing. The eukaryotic expression plasmid pIZT/V5-His (Invitrogen, Carlsbad, CA, USA) was used to express the CypD fusion protein with V5 and 6× His tags in insect cells (pIZT-CypD, Figure 4B). The pIZT/V5-His empty vector (pIZT, Figure 4B) served as a negative control. The specific CypD antibody was prepared by Bioworld Technology, Inc. (Nanjing, China).

### 2.9. Sequence Analysis

To determine the lysine site, the amino acid sequences of the CypD proteins from *Homo sapiens* (accession number: NP_005720), *Mus musculus* (accession number: NP_598845), and *S. litura* were obtained from GenBank and aligned using DNAMAN V6 software.

### 2.10. Site-Directed Mutation and the Construction of pIZT-CypDK125R Plasmid

To test the acetylation site of CypD, we constructed the mutation plasmid pIZT-CypDK125R for expressing the mutation CypDK125R protein using a Thermo Scientific Phusion Site-Directed Mutagenesis Kit (#F541; Thermo Fisher Scientific, Waltham, MA, USA) according to the manufacturer’s instructions. Briefly, a pair of 5′-phosphorylated primers (CypDK125R-F: 5′-P-GCTTGATGGCAGGCATGTGGTCT-3′; CypDK125R-R: 5′-P-CAGCTGGTCTTGACGCAGGTGA-3′) were designed using the pIZT-CypD plasmid as a template with the mutation site (Red) located in the middle of the forward primer. The pIZT-CypD plasmid was amplified by PCR with the two phosphorylated primers, and the products were linear fragments. The linear fragments were treated with a Dpn I enzyme to remove the template and then ligated into a circular plasmid with T4 DNA ligase. The mutation site was confirmed by sequencing.

### 2.11. Analysis of Apoptotic Cells Using Annexin V-FITC/PI

Analysis of apoptotic cells was performed using an annexin V-FITC/PI apoptosis detection kit (#A211-02; Vazyme) according to the manufacturer’s instructions. Briefly, Spli221 cells at 24 h post-transfection with 20 μL MbBV particles or hemocytes at 6 d p.p. were harvested and incubated with 400 μL FITC-binding buffer for 5 min at room temperature. Then, 5 μL FITC-conjugated antibody and 5 μL PI were added and incubated for 20 min on ice in darkness. An Olympus 71 inverted fluorescence microscope was used to detect apoptotic cells. Annexin V-FITC was used to identify the early apoptotic cells that showed green fluorescence; PI was used to identify the late apoptotic cells that showed red fluorescence.

### 2.12. Analysis of Apoptosis Using an Annexin V-PE Detection Kit

The analysis of apoptotic pIZT-CypD-transfected or pIZT-CypDK125R-transfected cells was performed using an annexin V-PE apoptosis detection kit (Beyotime, Shanghai, China) according to the manufacturer’s instructions. Briefly, the cells at 24 h post-transfection with 20 μL MbBV particles were collected and incubated with 195 μL of PE-binding buffer for 5 min at room temperature. Then, 5 μL of PE-conjugated antibody was added and incubated for 20 min on ice in the dark. An Olympus 71 inverted fluorescence microscope was used to detect apoptotic cells. An annexin V-PE antibody was used to identify apoptotic cells by showing red fluorescence.

### 2.13. Immunofluorescence

Immunofluorescence was performed as previously described [38] with minor modifications. Briefly, Spli221 cells were seeded in 12-well plates at a density of 1 × 10^5^ cells per well for transfection. The cells were washed with PBS and fixed for 15 min in 3.7% formaldehyde. CypD was identified using the CypD-specific antibody (1:2000; Bioworld Technology, Inc., Nanjing, China) and HRP-labeled goat anti-mouse IgG (H + L) secondary antibody (#A0216; Beyotime). Before imaging, the labeled cells were incubated with phalloidin (1:40; Sigma, Darmstadt, Germany) diluted in PBS for 1 h at 37 °C. Cells were washed with 1× PBS and incubated with 4′,6-diamidino-2-phenylindole (DAPI) (1: 1000; Roche) for 5 min. A 5-μL drop of fluoroshield with N-Tris (hydroxylmethyl) methyl-2-aminoethanesulfonic acid (TES) buffer (EMS Technical Data Sheets Cat.17985-30) was applied to the center of the well. Cells were imaged using an Olympus 71 inverted-fluorescence microscope.

### 2.14. Immunoprecipitation (IP) and Immunoblotting

Immunoprecipitation was performed using a protein A + G agarose kit (# P2012; Beyotime) according to the manufacturer’s instructions. Briefly, cells were lysed with radioimmunoprecipitation assay (RIPA) buffer (#P0013B; Beyotime). The prepared lysates were immunoprecipitated with anti-CypD antibodies (Bioworld Technology, Inc., Nanjing, China) at 4 °C for 12 h. Each experiment using this assay was performed at least three times independently. Immunoblotting was performed with the following primary antibodies: anti-Ac-Lys antibody, or anti-mouse V5 Tag monoclonal antibodies. A goat anti-mouse horseradish peroxidase-conjugated antibody was used as the secondary antibody.

### 2.15. Data Analysis

Data were analyzed using GraphPad Prism ver. 5 (GraphPad Inc., La Jolla, CA, USA). Statistical significance in experiments with two treatments was examined using an unpaired *t*-test, and the differences between three-group data were analyzed by one-way ANOVA. Semi-quantification of the bands was performed using ImageJ software ver. 1.49v (National Institutes of Health, Bethesda, MD, USA), with the intensity values normalized to the corresponding band of the internal reference; * *p* < 0.05, ** *p* < 0.01, and *** *p* < 0.001 were considered statistically significant. All of the data are presented as the mean ± SEM of at least three independent experiments.

## 3. Results

### 3.1. M. bicoloratus Parasitism Upregulates the Expression of CypD and Promotes CypD Acetylation in the Apoptotic Hemocytes of S. litura

To evaluate whether mitochondrial cyclophilin D (CypD) plays a role in the apoptotic response of *S. litura* hemocytes following *M. bicoloratus* parasitization in vivo, the protein levels of CypD were quantified by Western blot analysis after infection of *S. litura*. CypD was significantly upregulated in the hemocytes at 6 and 10 d p.p. (* *p* < 0.05; ** *p* < 0.05; p.p.: post-parasitized hemocytes; Figure 1A).

The anti-Ac-Lys antibody is commonly used for detecting the acetylation of lysine. To improve the specificity of CypD acetylation detection, we first isolated and enriched the CypD protein using the anti-CypD antibody by IP technology from the host hemocytes. Then, the acetylation testing of CypD was performed with the anti-Ac-Lys antibody. Parasitism promoted the acetylation of CypD in the host hemocytes at 6 and 10 d p.p. (Figure 1B).

To determine whether these changes were related to apoptotic processes in the parasitized hemocytes, annexin V-FITC/PI detection was used to measure apoptosis. Annexin V-FITC can bind to phosphatidylserine (PS) on the outer surface of the early apoptotic cells; PI, a nucleic acid dye, can penetrate the late apoptotic cell membrane and label the nucleus. The hemocytes from unparasitized larvae and parasitized larvae were compared for the proportion of apoptotic cells. There was a significantly higher proportion of apoptotic cells in the parasitized hemocyte population than in the unparasitized population (* *p* < 0.05; ** *p* < 0.01; *** *p* < 0.001; Figure 1C,D; green: early apoptosis; red: late apoptosis). These data were consistent with previous results [35] and suggested that *M. bicoloratus* parasitism promotes CypD expression and acetylation of host hemocytes. It also induces apoptosis that may be involved in the effects of CypD.

**Figure 1 viruses-15-01491-f001:**
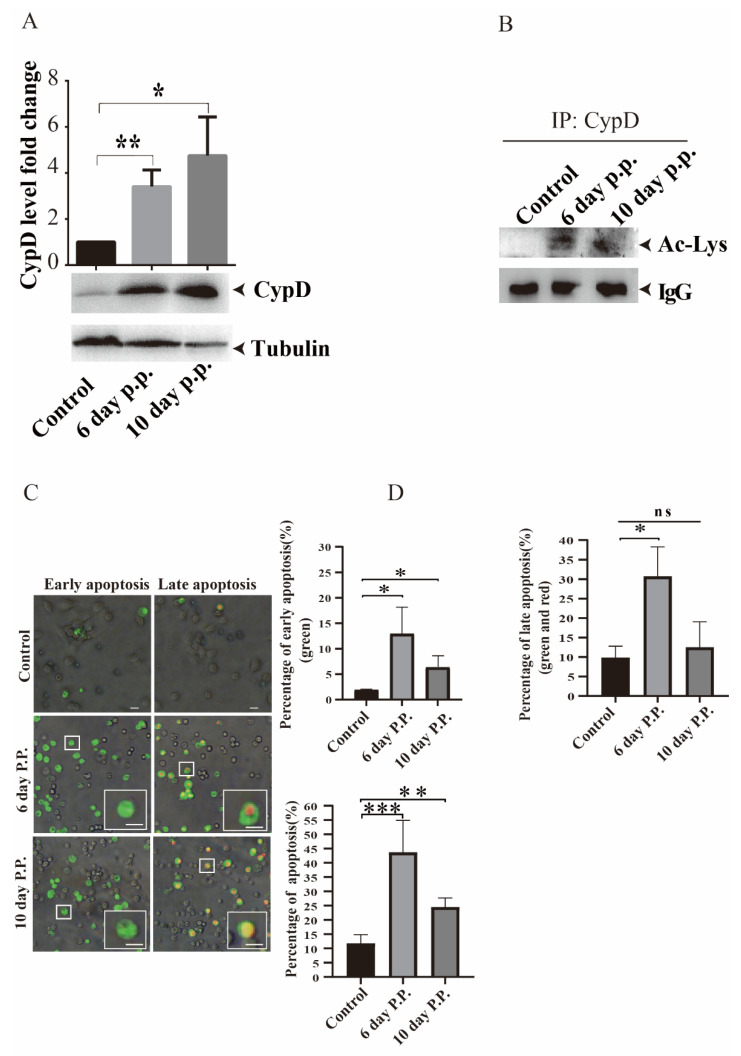
*Microplitis bicoloratus* parasitism upregulates the expression of CypD and promotes CypD acetylation in the apoptotic hemocytes of *Spodoptera litura*. (**A**) The results of semi-quantitative analysis of Western blot signals showed a significant increase in CypD expression in hemocytes at 6 or 10 d p.p. (**B**) Cell lysates of host hemocytes were analyzed by immunoprecipitation (IP) with anti-CypD antibodies and immunoblotting with anti-Ac-Lys antibody. (**C**) Apoptotic hemocytes induced by *M. bicoloratus* parasitism stained with annexin V-FITC/PI and observed under an inverted fluorescence microscope. The samples were analyzed for green fluorescence (FITC) and red fluorescence (PI). Different labeling patterns of the apoptotic hemocytes: early apoptotic cells, annexin V-FITC positive and PI negative, and late apoptotic cells, both annexin V-FITC and PI positive. The white squares in both 6 day p.p. and 10 day p.p. showed that the representative cells were in the early or late stages of apoptosis. Scale bar, 20 μm. (**D**) Comparison of the percentages of apoptotic cells between NP and p.p. hemocytes. NP: non-parasitized hemocytes (control); p.p.: post-parasitized hemocytes. Data are presented as mean ± SEM of three independent experiments. Significant differences are indicated by asterisks (annexin V-FITC: labels phosphatidylserine, PS; PI: labels nucleus; * *p* < 0.05, ** *p* < 0.01, *** *p* < 0.001, two-tailed Student’s *t*-test).

### 3.2. CypD Expression and Acetylation Are Significantly Upregulated in Spli221 Cells Undergoing MbBV-Induced Apoptosis

Parasitism promotes CypD expression and acetylation of host hemocytes. To determine whether CypD expression and acetylation are associated with the apoptosis induced by MbBV carried by *M. bicoloratus*, we purified MbBV particles from the ovaries of *M. bicoloratus* females and used them to infect Spli221 cells, derived from the pupal ovaries of *S. litura*. After a 24 h viral induction period, the infected Spli221 cells were significantly more apoptotic than the uninfected controls (* *p* < 0.05; ** *p* < 0.01 Figure 2A,B). Western blot analysis confirmed that CypD expression and acetylation were upregulated in these apoptotic cells (* *p* < 0.05; ** *p* < 0.01; Figure 2C–E). As an integrator of mitochondrial function, CypD acetylation may cause the irreversible opening of mPTP and culminate in the loss of mitochondrial ∆Ψm [16,39]. We used a JC-1 fluorescent probe to examine the mitochondrial ∆Ψm and found that the MbBV-infection significantly decreased ∆Ψm in MbBV-infected Spli221cells compared with the uninfected cells (Figure 2F,G, * *p* < 0.05). These data suggest that CypD acetylation may be involved in MbBV-induced apoptosis.

### 3.3. Inhibition of CypD Acetylation Significantly Rescued the Apoptotic Cells Induced by MbBV

To study the role of CypD acetylation in cell apoptosis induced by MbBV, we used CsA and polydatin to inhibit CypD acetylation and observed the inhibition effect on MbBV-infected cells. To determine the efficiency of cyclosporine A (CsA) or polydatin on CypD deacetylation in Spli221 cells, IP assays and Western blot analyses were carried out to test the CypD acetylation levels in the CsA- or polydatin-treated cells at 24 h post-MbBV infection, respectively. The results showed that CypD acetylation levels in CsA- or polydatin-treated cells decreased compared with the untreated cells (Figure 3A). These results suggest that CsA and polydatin can suppress CypD acetylation.

To determine how lowered acetylation levels of CypD affect MbBV-mediated apoptosis, Spli221 cells 24 h post-treated with CsA or polydatin were infected with MbBV for a 24 h incubation period before measuring the levels of apoptosis through annexin V-FITC and PI detection. The percent of apoptotic cells significantly decreased in the cells treated with CsA or polydatin (* *p* < 0.05; ** *p* < 0.01; Figure 3B,C). These results indicate that CypD acetylation plays an important role in MbBV-mediated apoptosis.

### 3.4. Cloning of S. litura CypD and the Construction of pIZT-CypDK125R Plasmid

To test the acetylation site of *S. litura* CypD, a cDNA clone encoding the 495 bp *CypD* gene from the hemocytes of *S. litura* parasitized by *M. bicoloratus* was generated. The ORF of the clone encoded a 164 amino acid sequence (Figure 4A). We aligned *H. sapiens* (accession number: NP_005720), *M. musculus* (accession number: NP_598845), and *S. litura* CypD sequences and searched for homologous sites of acetylation. The corresponding lysine sites of *H. sapiens*, *M. musculus*, and *S. litura* were K167, K166, and K125, respectively (Figure 4A, red box). The acetylation site of mouse CypD is K166 [29]. Therefore, K125 of *S. litura* CypD was assumed to be the acetylation site. To verify this, we constructed the eukaryotic expression plasmid pIZT-CypD. We then performed site-directed mutagenesis of *S. litura* CypD using pIZT-CypD as a template and mutated K125 to R125 (pIZT-CypDK125R) (Figure 4B). Sequencing demonstrated that a mutant plasmid (pIZT-CypDK125R) was successfully constructed (Figure 4C).

### 3.5. MbBV Promoted CypD Activation by Acetylating CypD K125

To determine the acetylation site of *S. litura* CypD, we transfected the pIZT-CypD plasmid and the pIZT-CypDK125R plasmid into Spli221 cells. At 72 h post-transfection, the CypD-V5 and CypDK125R-V5 fusion proteins were detected in the cytoplasm of Spli221 cells by immunofluorescence (Figure 5A, red). CypD-V5 and CypDK125R-V5 proteins were also detected in the Spli221 cell lysate by Western blotting using an anti-V5 antibody (Figure 5B). These data revealed that both of the fusion proteins were normally expressed and distributed in the cytoplasm. To confirm that MbBV promotes acetylation of the CypDK125 site, we used MbBV to infect the Spli221 cells that were post-transfected with pIZT-CypD or pIZT-CypDK125R. At 24 h post-infection, immunoblotting analysis was performed on the lysates from the infected cells. The immunoblotting results showed that the CypD-V5 fusion protein underwent acetylation, while the CypDK125R-V5 fusion protein underwent no acetylation during MbBV infection, indicating that MbBV promoted the acetylation of CypDK125 (Figure 5C). To further determine the effect of this acetylation on MbBV-mediated apoptosis, we measured the levels of apoptosis by annexin V-PE detection. As expected, there was a significantly lower proportion of apoptotic cells in the pIZT-CypDK125R-transfected cell population than in the pIZT-CypD-transfected population during MbBV infection (*** *p* < 0.001; Figure 5D). These results suggest that MbBV promotes CypD activation by acetylating CypD K125 during MbBV-mediated cell apoptosis.

## 4. Discussion

Insects respond to the invasion of pathogenic microorganisms such as viruses by activating their innate immune response to maintain body homeostasis. However, some viruses have developed mechanisms to promote their survival by inhibiting the host immune response. *M. bicoloratus* parasitism regulates *S. litura* hemocyte apoptosis, resulting in immunosuppression [33,34,40]. *M. bicoloratus* parasitism promotes CypD–p53 interaction to induce host hemocyte apoptosis [35]. In the present study, we examined the expression and acetylation of CypD in host hemocytes or MbBV-infected cells. Parasitism of *M. bicoloratus* or MbBV infection promoted the expression and acetylation of CypD. To test the correlation between CypD acetylation and MbBV-induced apoptosis, we used CsA and polydatin to inhibit the acetylation of CypD. We found that the acetylation inhibition significantly rescued the MbBV-induced apoptosis. We used point mutation technology to show that the deacetylation of CypDK125R occurs in MbBV-infected cells, and we also rescued the apoptotic cells induced by MbBV. These findings illustrate the important role of MbBV-mediated CypD acetylation in the insect immunosuppressive process.

Natural parasitism of *M. bicoloratus* stimulates CypD acetylation of host hemocytes and promotes the occurrence of apoptosis (Figure 1B–D). However, we cannot determine whether MbBV causes these phenomena. To address this issue, we purified MbBV particles from the calyces of female *M. bicoloratus* and infected Spli221 cells, a cell line derived from the pupal ovary of *S. litura.* MbBV infection resulted in CypD acetylation in the infected cells (Figure 2D,E), accompanied by increased expression of CypD and apoptosis (Figure 2A–C). These results suggest that the activation of CypD may occur through acetylation mediated by MbBV during the parasitism of *S. litura* by *M. bicoloratus*. We previously found, using whole-genome sequencing of MbBV, that MbBV contains 116 genes [7]. Parasitized *S. litura* larvae infected with MbBV had 2441 genes downregulated and 299 genes upregulated [34]. In the present study, we confirmed that parasitism or MbBV infection promotes CypD expression and acetylation, but we do not know which MbBV gene is responsible for this. This information will require more research to clarify how MbBV genes hijack host CypD.

CypD is also an activator of the mitochondrial permeability transition pore (mPTP). Sustained opening of the mPTP results in loss of membrane potential, uncoupling of oxidative phosphorylation, ATP depletion, and increased production of reactive oxygen species. These effects will ultimately lead to cell death [16,39,41]. We also used a JC-1 fluorescent probe to examine ∆Ψm of mitochondria and found that MbBV infection significantly decreased ∆Ψm of infected insect cells compared with the uninfected cells (Figure 2F,G). These data suggest that the mitochondrial membrane was damaged and the mPTP opened. We also observed that CypD can interact with ANT1 in Spli221 cells and cytochrome c (Cyt c) was released from the hemocyte mitochondria during *M. bicoloratus* parasitism (unpublished data). These data suggest that CypD involves a mitochondrial pathway in cell apoptosis during immunosuppression mediated by MbBV.

The anti-Ac-Lys antibody is used to detect the acetylation of protein lysine residues. During parasitism or MbBV infection, many other proteins may undergo acetylation in addition to CypD. Therefore, if we directly utilized the antibody alone to detect the acetylation of CypD, it may cause significant errors. To solve this problem, we combined IP technology and acetylation detection. Using the anti-CypD antibody, we first enriched CypD proteins from the host hemocytes or MbBV-infected cells by IP technology. We then utilized the anti-Ac-Lys antibody to check the acetylation level of CypD proteins. This allowed us to improve the specificity of detecting CypD acetylation and reduce errors.

Increased levels of CypD acetylation in MbBV-mediated apoptotic cells (Figure 2D,E) do not indicate that CypD acetylation is required for MbBV-mediated apoptosis. To study CypD acetylation in MbBV-induced apoptosis, we needed to inhibit CypD acetylation to observe its impact on apoptosis. We chose CsA and polydatin to suppress the acetylation of CypD. CsA, an immunosuppressive agent, binds to cyclophilin and inhibits cyclophilin activity [42]. The acetylation site of CypD is close to its CsA-binding domain, which suggests that CsA may affect CypD acetylation modification [25]. Polydatin can deacetylate CypD by activating SIRT1 and SIRT3 [43,44,45,46]. In the present study, we found that CsA or polydatin can effectively inhibit CypD acetylation mediated by MbBV (Figure 3A). The acetylation inhibition rescues the apoptotic cells induced by MbBV (Figure 3B,C). These data demonstrate that CypD acetylation is required for MbBV-induced apoptosis.

Lys acetylation is a reversible post-translational modification that is highly conserved from prokaryotes to humans and plays an important role in cellular functions such as RNA splicing, DNA damage repair, cell cycle, and nuclear transport [20,47]. Because proteins typically contain multiple lysine sites, it is difficult to identify the acetylation site of CypD. We aligned human, mouse, and *S. litura* CypD amino acid sequences and found that human K167, mouse K166, and *S. litura* K125 are homologous lysine sites (Figure 4A). Hafner et al. suggested that the acetylation site of mouse CypD, under stress conditions, is K166 and found that the acetylation site of CypD is highly conserved [29,47]. It is tempting to speculate that *S. litura* K125 is also an acetylation site under virus infection. K125-point mutation and acetylation detection results showed that the CypD-V5 fusion protein undergoes acetylation, while the CypDK125R-V5 protein does not undergo acetylation in MbBV-infected cells (Figure 5B,C). These results indicate that *S. litura* CypD is acetylated on K125 and also suggest that *S. litura* CypD has only one acetylation site. On the other hand, apoptosis detection results reveal that the drop in apoptosis upon lack of K125 acetylation is only 40% (Figure 5D), suggesting that MbBV may promote cell apoptosis through a variety of mechanisms, with CypD acetylation being just one of these channels.

The natural immunity of insects is divided into humoral immunity and cellular immunity, which both have differences and connections and jointly recognize and eliminate pathogens [48,49]. Cellular immunity clears pathogens through hemocyte-mediated phagocytosis, nodulation, and encapsulation. Humoral immunity mainly works by inducing the production of antimicrobial peptides and other effector molecules through signal transduction and immune pathways. This is closely related to coagulation, melanosis, and reactive oxygen species (ROS) production [48]. In the parasitoid/polydnavirus/host system, our findings did not reveal how MbBV regulates host humoral immunity. However, they confirmed that MbBV can induce apoptosis in host hemocytes by promoting CypD acetylation and weakening the host immune response. We have also observed that MbBV can downregulate the expression of attacin, an antimicrobial peptide (unpublished data). The parasitoid wasps may also be involved in suppressing host immune responses. Zhang et al. showed that a Cu/Zn superoxide was secreted by the immature parasitoid to protect against ROS generated by host hemocytes in response to MbBV during development [50]. These studies suggest that the immune confrontation between parasitoids and hosts is a complex process, and its molecular mechanism remains undetermined.

## 5. Conclusions

Parasitism can trigger an immune response in the insect host. We demonstrated that MbBV can regulate CypD acetylation of host hemocytes to promote apoptosis that suppresses host cellular immunity. The acetylation of CypD may alter mitochondrial permeability, resulting in the release of apoptotic proteins. We found that the only acetylation site of *S. litura* CypD is on K125. These data establish a causal link between innate immune suppression and CypD acetylation and identify the important role of CypD acetylation in the insect immunosuppressive process. Given the high conservation of Lys acetylation from prokaryotes to humans, a similar immune mechanism may exist in many other species. Nevertheless, there exist some limitations in our study. The results cannot more accurately reflect the situation within the organism, for instance, because we did not per-form the K125 point mutation or knock out the CypD gene in the genome.

## Figures and Tables

**Figure 2 viruses-15-01491-f002:**
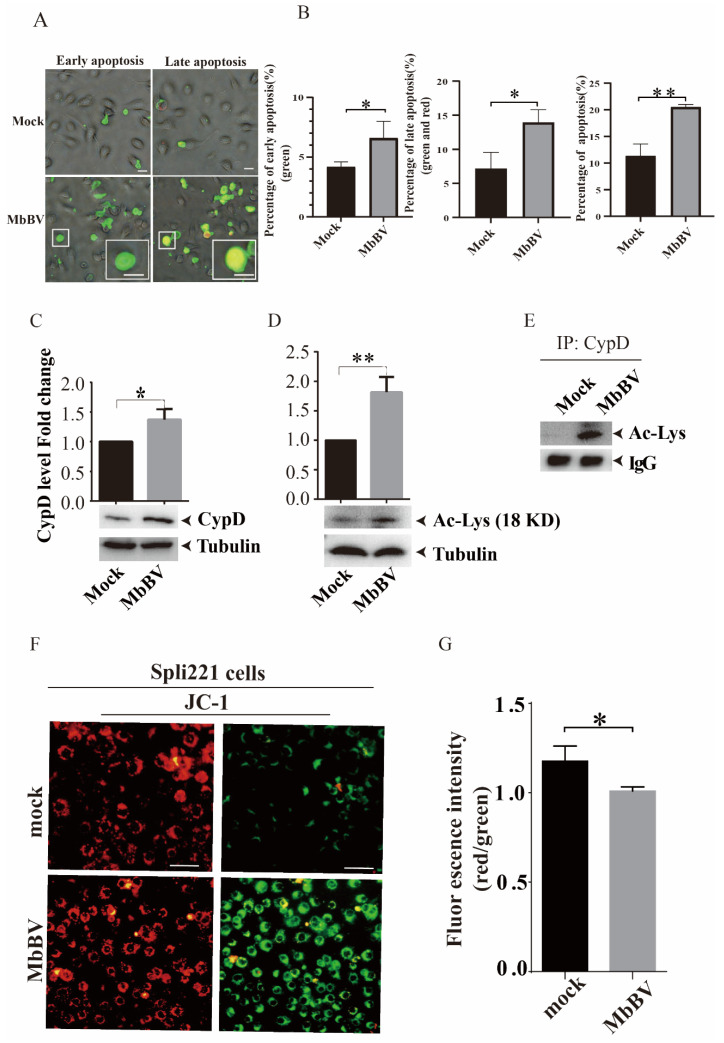
CypD expression and acetylation are significantly upregulated in Spli221 cells undergoing MbBV-induced apoptosis. (**A**) Representative images and (**B**) analysis of apoptotic cells demonstrating that Spli221 cells infected by MbBV had significantly higher levels of apoptosis compared with non-infected cells. Inverted fluorescence microscopy shows cells in the early stages of apoptosis labeled with annexin V-FITC (green) and cells in the late stages of apoptosis labeled with PI (red; scale bar, 20 μm). The white squares in MbBV showed that the representative cells were in the early or late stages of apoptosis. (**C**) Western blot and semi-quantification analyses showed a significant increase in CypD protein in MbBV-infected Spli221 cells. * *p* < 0.05. (**D**,**E**) Immunoblotting and immunoprecipitation analyses showed a significant increase in CypD acetylation in MbBV-infected Spli221 cells. ** *p* < 0.01. Error bars represent SEM. Data represent three independent experiments. (**F**) Representative fluorescence microscope images and (**G**) quantitative analysis of the mitochondrial membrane potential (∆Ψm) showed that ∆Ψm was significantly reduced in MbBV-infected Spli221 cells. The normal ∆Ψm is labeled with J-aggregates (red), and the decreased ∆Ψm is labeled with J-monomer (green).

**Figure 3 viruses-15-01491-f003:**
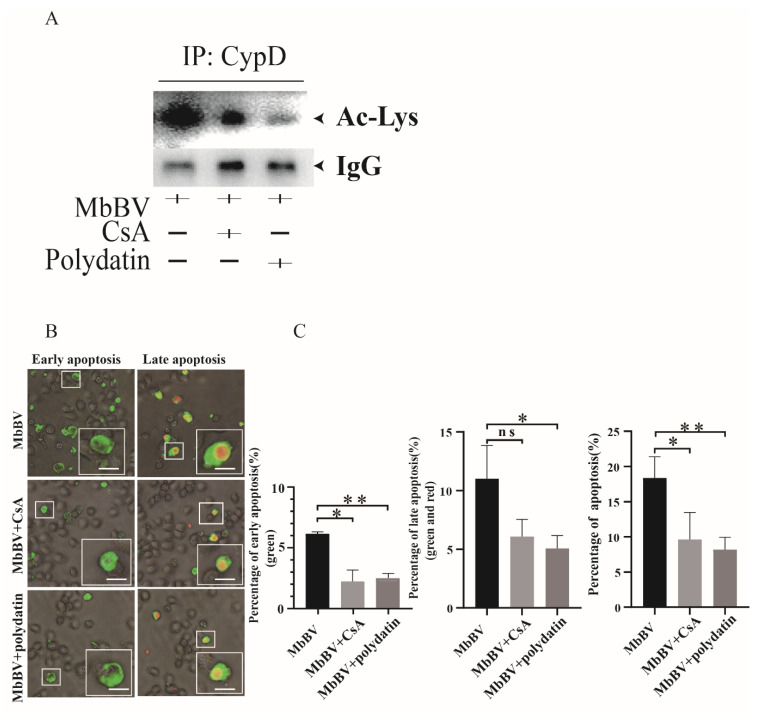
Inhibition of CypD acetylation significantly rescued the apoptotic cells induced by MbBV. (**A**) Anti-Ac-Lys IB analysis of IP with anti-CypD antibody in the Cell lysates of CsA-treated or polydatin-treated Spli221 cells infected with MbBV for 24 h. (**B**) Representative images and (**C**) analysis of apoptotic cells demonstrating that the proportion of apoptotic cells was significantly reduced in CsA-treated or polydatin-treated Spli221 cells infected with MbBV compared with untreated cells. Inverted fluorescence microscopy shows cells in the early stages of apoptosis labeled with annexin V-FITC (green) and cells in the late stages of apoptosis labeled with PI (red). The white squares showed that the representative cells were in the early or late stages of apoptosis. * *p* < 0.05, ** *p* < 0.01. Error bars represent standard error of mean (SEM). Data represent three independent experiments.

**Figure 4 viruses-15-01491-f004:**
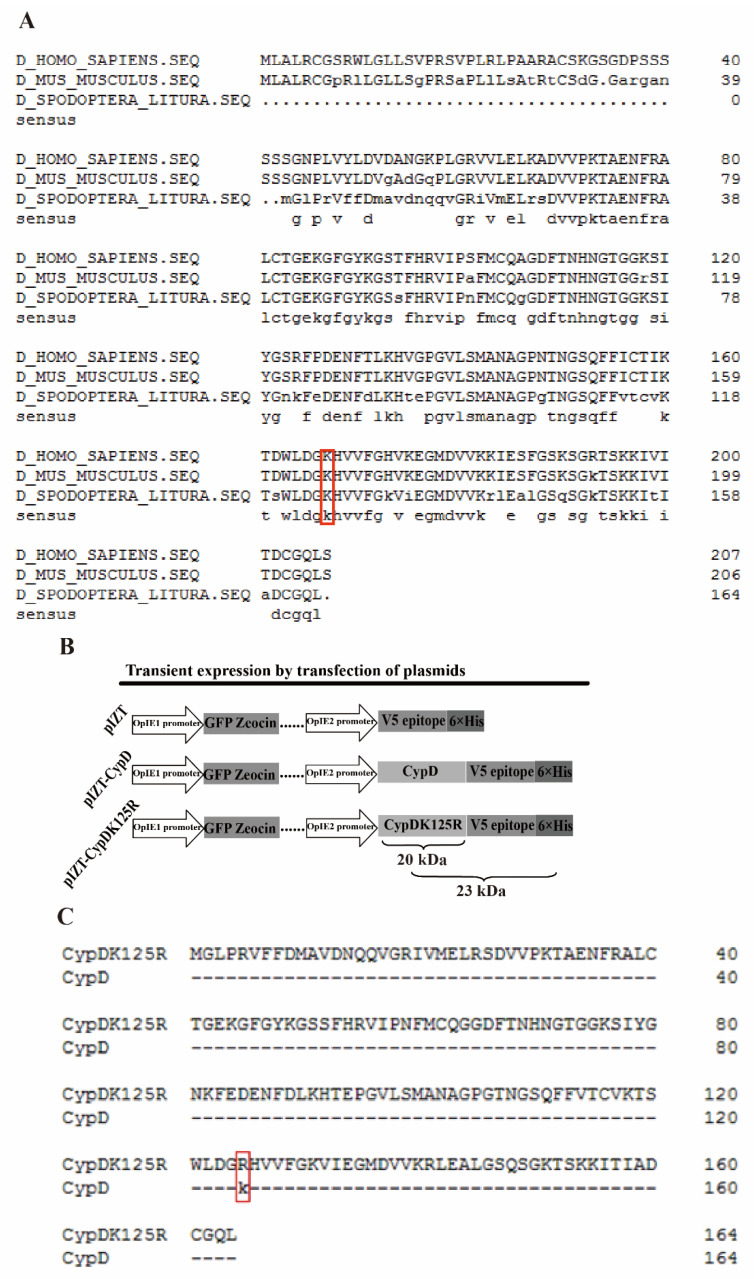
Cloning of *S. litura* CypD and the construction of pIZT-CypDK125R plasmid. (**A**) Sequence homology analysis of cyclophilin D from *Homo sapiens*, *Mus musculus*, and *S. litura*. The homologous sites of lysine acetylation are shown in red squares. (**B**) CypD plasmid construct containing the OpIE1 promoter for expression of GFP-Zeocin fusion proteins and the OpIE2 promoter for expression of V5-fusion proteins using the pIZT/V5-His vector. (**C**) The sequencing results of the mutant plasmid pIZT-CypDK125R, and the K125 was changed to R125 (red square).

**Figure 5 viruses-15-01491-f005:**
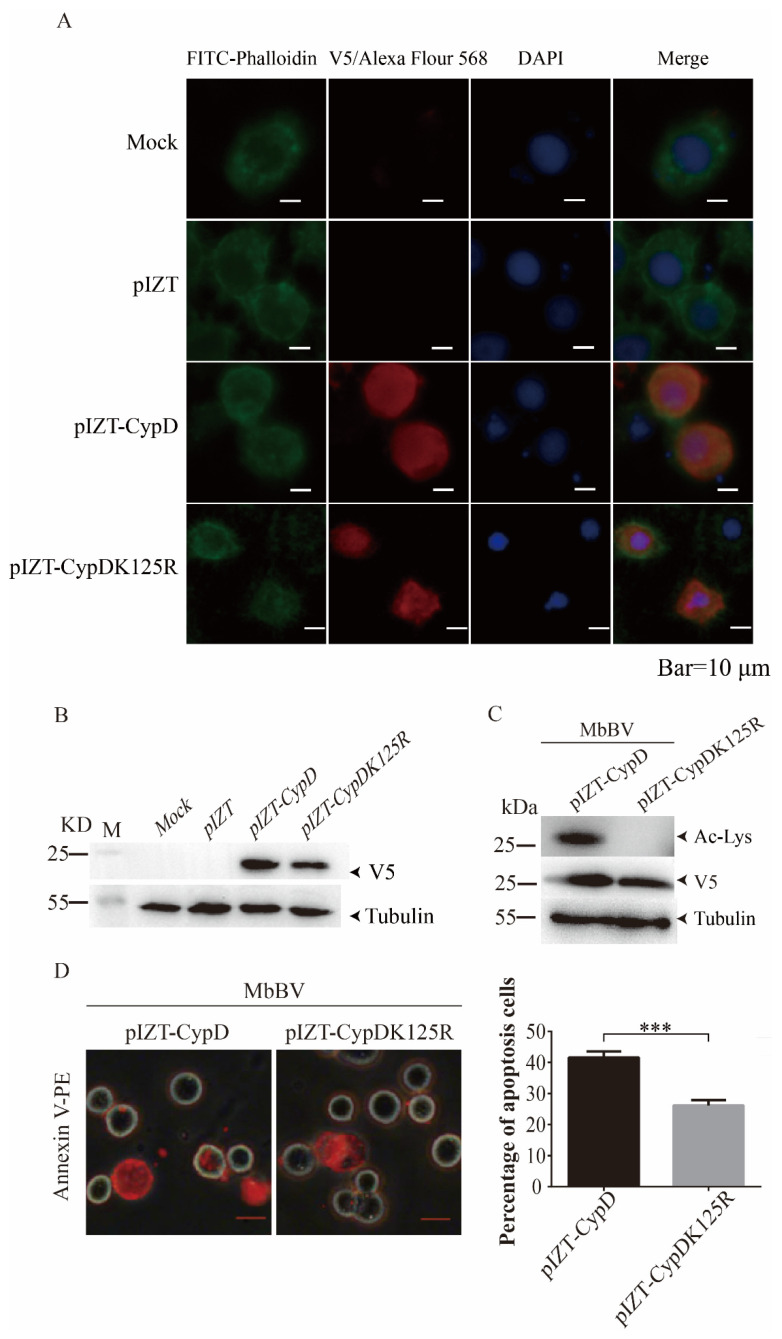
MbBV promoted CypD activation by acetylating CypD K125. (**A**) CypD-V5 and CypDK125R-V5 fusion proteins were detected in the cytoplasm of Spli221 cells by inverted fluorescence microscopy (red, scale bar, 10 μm). (**B**) The 18-kDa CypD-V5 and CypDK125R-V5 fusion protein was detected in Spli221 cell lysates at 72 h post-transfection. (**C**) Immunoblotting analysis showed the deacetylation of CypDK125R-V5 fusion protein in MbBV-infected Spli221 cells. (**D**) Analysis of apoptotic cells demonstrating that the proportion of apoptotic cells was significantly reduced in pIZT-CypDK125R-transfected cells infected with MbBV compared with pIZT-CypD-transfected cells. Inverted fluorescence microscopy shows apoptotic cells labeled with annexin V-PE (red). *** *p* < 0.001. Error bars represent standard error of mean (SEM). Data represent three independent experiments.

## Data Availability

All of the data relevant to this study can be found in this article.

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
