# Peer review of "Microplitis bicoloratus Bracovirus Promotes Cyclophilin D-Acetylation at Lysine 125 That Correlates with Apoptosis during Insect Immunosuppression"

_viruses, 2023, doi:10.3390/v15071491_

Round 1
Reviewer 1 Report
This manuscript is well design and well presented. I think it is good enough to be published.
The quality of English is good.
Author Response
Dear reviewer,
Thank you very much for your advice and affirmation of the paper. Your suggestions have been read and we understand their meaning. We will improve this manuscript and submit it as soon as possible.
Thank you and best regards,
your sincerely,
Ming Li, PhD
Reviewer 2 Report
The study methods are valid and reliable. There are enough details provided in order to replicate the study. The data is presented in an appropriate way. The text in the results add to the data and it is not repetitive. Results are discussed from different angles and placed into context without being overinterpreted.
The conclusions answer the aim of the study. The conclusions are supported by references and own results.
Specific comments on weaknesses of the article and what could be improved:
Major points - none
Minor points
1. Please, state the limitations of the study
2. Could you please discuss the clinical implications of the results
Author Response
Dear reviewer,
Thanks very much for your kind work and consideration on publication of our paper. On behalf of my co-authors, we would like to express our great appreciation to you. Your suggestions have been read and we understand their meaning. Discuss ​these suggestions By DAN YU and PAN ZHANG, we explain your opinions as followings :
- Please, state the limitations of the study.
Response: Add “Nevertheless, there exist some limitations in our study. The results cannot more accurately reflect the situation within the organism, for instance, because,we did not per-form the K125 point mutation or knock out CypD gene in the genome” in the conclusion section.
- Could you please discuss the clinical implications of the results
Response: due to our research on insect innate immunity, it is somewhat difficult to discuss the clinical implications of the results. Therefore, we will not conduct this discussion temporarily. We hope you can understand that.
Thank you and best regards.
Yours sincerely,
Ming Li, PhD
Reviewer 3 Report
The work submitted by Dan Yu and colleagues reports the correlation between the upregulation of Cyclophilin D in mitochondria of Spodoptera litura larvae infected by theparasitoid wasp Microplitis bicoloratus, its acetylation at lysine 125 and an increases in mitochondria apoptosis upon a drop in the membrane potential across the inner mitochondrial membrane.
While the data are in line with previous observations in other systems and in support of the main conclusions, hence no further experimental work is required, the manuscript would benefit from some editing.
Comments:
Title:
better “ Microplitis bicoloratus bracovirus promotes cyclophilin D-acetylation at lysine 125 that correlates with apoptosis during insect immunosuppression” – the data show only a correlation
General comment: please go through the manuscript with a lay reader in mind and make more use of metacomentaries (see https://www.ncbi.nlm.nih.gov/pmc/articles/PMC9997112/)
Introduction:
L32: give the reader here more background on Spodoptera litura and why it is important to know about the infection of this nocturnal moth.
L42-43 is the mPTP pore in the inner membrane only or does it reach across both membrannes?
Results:
L218: please give the full name of CypD here again and say mitochondrial CypD (please do not assume that the reader switches permanently between the Result & Method section/Introduction to find out what the abbreviations mean)
L220 …by Western blot analysis after infection of S. litura.
L221: give the full name of p.p. here again
Figure 1A the label at the bottom row of the Western blot is missing
Figure 1D & Figure 2B & Figure 3B: please use the same scale on the y-axis for all three graphs as the first two are summed up in the third one. Alternatively, you present only one bar graph with different colours for early and late apoptosis in the same bar
Legend Figure 1: please give the full name of p.p.; please explain what FITC and PI label.
L 243 please explain what FITC and PI label.
L 256: introduce Spli221 cells to the reader here
Figure 2C: please use the same scale on the y-axis for both graphs
L 282: please give the full name of CsA here
Figure 5A: there seems to be a change in localisation of overexpressed CypDK125R here, please comment (reduced nuclear staining?)
Discussion:
The Discussion is more a justification of the rational behind the experiments, better to comment on (i) the overlap with similar data in other systems (Ref 28, Ref 39) and why this might be so highly conserved across species, (ii) why the drop in apoptosis upon lack of K125 aceytylation is only 40%, (iii) why the cellular localisation of the K125R mutant appears to change and (iv) why the data matter with regard to the S. litura that is a serious polyphagous pest in Asia.
The english is fine, only minor editing is required. However, the text is difficut to follow and should be edited to make it easier for educated lay readers to follow the story. Please make more use of metacommentaries (see: https://www.ncbi.nlm.nih.gov/pmc/articles/PMC9997112/) and story telling (see https://www.advancedsciencenews.com/the-role-of-narrative-in-science/)
Author Response
Dear reviewer,
Thank you very much for your kind work. Your suggestions have been read and we understand their meaning. Discuss ​these suggestions By DAN YU and PAN ZHANG, we explain your opinions as followings :
Title: better “Microplitis bicoloratus bracovirus promotes cyclophilin D-acetylation at lysine 125 that correlates with apoptosis during insect immunosuppression” – the data show only a correlation.
Response: Change the title to “Microplitis bicoloratus bracovirus promotes cyclophilin D-acetylation at lysine 125 that correlates with apoptosis during insect immunosuppression”.
Introduction:
L32: give the reader here more background on Spodoptera litura and why it is important to know about the infection of this nocturnal moth.
Response: Add “a serious agricultural pest” in L32.
L42-43 is the mPTP pore in the inner membrane only or does it reach across both membrannes?
Response: Add “reaches arcoss both the outer membrane and the inner membrane” in L42-43.
Results:
L218: please give the full name of CypD here again and say mitochondrial CypD (please do not assume that the reader switches permanently between the Result & Method section/Introduction to find out what the abbreviations mean)
Response: Add “mitochondrial cyclophilin D (CypD)” in L218.
L220 …by Western blot analysis after infection of S. litura.
Response: Add “after infection of S. litura” in L220.
L221: give the full name of p.p. here again
Response: Add “p.p.: post-parasitized hemocytes” in L221.
Figure 1A the label at the bottom row of the Western blot is missing
Response: Add “tubulin” in Figure 1A.
Figure 1D & Figure 2B & Figure 3B: please use the same scale on the y-axis for all three graphs as the first two are summed up in the third one. Alternatively, you present only one bar graph with different colours for early and late apoptosis in the same bar
Response: The adjustment of the same scale on the y-axis has been completed.
Legend Figure 1: please give the full name of p.p.; please explain what FITC and PI label.
Response: Add “p.p.: post-parasitized hemocytes” and “annexin V-FITC: labels phosphotidylserine, PS; PI: labels nucleus” in legend Figure 1.
L 243 please explain what FITC and PI label.
Response: Add “Annexin V-FITC can bind to phosphotidylserine (PS) on the outer surface of the early apoptotic cells; PI, a nucleic acid dye, can penetrate the late apoptotic cell membrane and label the nucleus” in L243.
L 256: introduce Spli221 cells to the reader here
Response: Add “derived from the pupal ovaries of S. litura” in L 256.
Figure 2B: please use the same scale on the y-axis for both graphs
Response: The figure adjustment has been completed.
L 282: please give the full name of CsA here
Response: Add “cyclosporine A (CsA)” in L 282.
Figure 5A: there seems to be a change in localisation of overexpressed CypDK125R here, please comment (reduced nuclear staining?)
Response: Our observation shows that there seems no significant change in localisation of overexpressed CypD and CypDK125R proteins, and although the displayed image shows a reduced nuclear staining, it may be due to the relatively small morphology of the CypDK125R cells we accidentally selected. Of course, reduced nuclear staining may also be due to the role of CypDk125R, but more research is needed to confirm this. If overexpressed CypDK125R can reduce the nuclear, which may mean that the deacetylation of CypD still has unknown functional activity. These speculations require further research
Discussion:
The Discussion is more a justification of the rational behind the experiments, better to comment on (i) the overlap with similar data in other systems (Ref 28, Ref 39) and why this might be so highly conserved across species, (ii) why the drop in apoptosis upon lack of K125 aceytylation is only 40%, (iii) why the cellular localisation of the K125R mutant appears to change and (iv) why the data matter with regard to the S. litura that is a serious polyphagous pest in Asia.
Response: Thanks the reviewer’s suggestions. To maintain the overall framework of the discussion section, we add “On the other hand, apoptosis detection results reveal that the drop in apoptosis upon lack of K125 acetylation is only 40% (Figure 5D), suggesting that MbBV may promote cell apoptosis through a variety of mechanism, with CypD acetylation being just one of these channels” in line 434.
Comments on the Quality of English Language
The english is fine, only minor editing is required. However, the text is difficut to follow and should be edited to make it easier for educated lay readers to follow the story. Please make more use of metacommentaries (see: https://www.ncbi.nlm.nih.gov/pmc/articles/PMC9997112/) and story telling (see https://www.advancedsciencenews.com/the-role-of-narrative-in-science/)
Response: Worried about the time available, we are not revising the paper extensively. Thanks your suggestions very much.
Thank you and best regards.
Yours sincerely,
Ming Li, PhD